# Population-specific positive selection on low CR1 expression in malaria-endemic regions

**Paolo Alberto Lorenzini**[1,2☯], **Elena S. Gusareva**[1,2,3☯], **Amit Gourav Ghosh**[1,2], **Nurul Adilah Binte Ramli**[1,2], **Peter Rainer Preiser**[4], **Hie Lim Kim**[1,2,3]*

**1** Asian School of the Environment, Nanyang Technological University, Singapore, Singapore, **2** The GenomeAsia 100K Consortium, Singapore, Singapore, **3** Singapore Centre for Environmental Life Sciences Engineering, Nanyang Technological University, Singapore, Singapore, **4** School of Biological Sciences, Nanyang Technological University, Singapore, Singapore

☯ These authors contributed equally to this work.

* HLKIM@ntu.edu.sg

**Data Availability Statement:** The datasets analyzed for this study can be found in the European Genome-phenome Archive (EGA) under accession number EGAS00001002921.

## Abstract

Complement Receptor Type 1 (*CR1*) is a malaria-associated gene that encodes a transmembrane receptor of erythrocytes and is crucial for malaria parasite invasion. The expression of CR1 contributes to the rosetting of erythrocytes in the brain bloodstream, causing cerebral malaria, the most severe form of the disease. Here, we study the history of adaptation against malaria by analyzing selection signals in the *CR1* gene. We used whole-genome sequencing datasets of 907 healthy individuals from malaria-endemic and non-endemic populations. We detected robust positive selection in populations from the hyper-endemic regions of East India and Papua New Guinea. Importantly, we identified a new adaptive variant, rs12034598, which is associated with a slower rate of erythrocyte sedimentation and is linked with a variant associated with low levels of CR1 expression. The combination of the variants likely drives natural selection. In addition, we identified a variant rs3886100 under positive selection in West Africans, which is also related to a low level of CR1 expression in the brain. Our study shows the fine-resolution history of positive selection in the *CR1* gene and suggests a population-specific history of *CR1* adaptation to malaria. Notably, our novel approach using population genomic analyses allows the identification of protective variants that reduce the risk of malaria infection without the need for patient samples or malaria individual medical records. Our findings contribute to understanding of human adaptation against cerebral malaria.

## Introduction

CR1 is a transmembrane glycoprotein expressed on the surface of peripheral blood cells. This immune-regulatory cellular receptor clears external pathogens and damaged cells from the human body through complement activation [1–4]. In addition, CR1 is a well-known host receptor hijacked by malaria parasites (*Plasmodium falciparum*) to invade red blood cells (RBCs) during the blood stage of the malaria life cycle [5, 6]. The binding of CR1 to the *Plasmodium falciparum* Erythrocyte Membrane Protein 1 (PfEMP1) ligand (S1 Fig) can induce

**Funding:** This research was supported by the Singapore Ministry of Education, Academic Research Fund Tier 1 (grant number 2017-T1-001-046 and grant number RG100/20). The funders had no role in study design, data collection and analysis, decision to publish, or preparation of the manuscript.

**Competing interests:** The authors have declared that no competing interests exist.

the aggregation of uninfected RBCs with infected cells, a phenomenon called 'rosetting' [7, 8]. Multiple rosettes cause the blockage of small blood vessels in the brain [9, 10], known as cerebral malaria. In fact, most malaria deaths amongst children in Africa are due to cerebral malaria [11, 12]. The pathogenesis of this severe form of malaria remains poorly understood, and treatment options are lacking [5, 7, 8].

Alleles related to the expression level of the *CR1* gene have been identified [7, 13–22]. Two alleles of the *CR1* gene, designated as the high (H) and low (L) alleles, cause a 10-fold difference in its expression level on the RBC surface [23]. The H and L alleles contain two non-synonymous variants rs2274567 [A > G] and rs3811381 [C > G], and an intronic variant rs11118133 [A > T] [21, 23–25] (S1 Fig). The H allele includes A/C/A nucleotides of the three variants while the L allele includes G/G/T nucleotides, respectively.

A protective effect of low *CR1* expression against malaria has been suggested. Homozygotes for the L allele (L/L) of the two variants rs2274567 and rs381131 showed significant protection against malaria in the hyperendemic region of Odisha in East India [13], whilst homozygotes for the H allele (H/H) of the two variants were associated with an increased risk of developing cerebral malaria in a population from the same area [17]. The heterozygote (L/H) of rs2274567 was associated with intermediate *CR1* expression levels and protection against severe malaria in populations from the highly endemic region of Papua New Guinea [20]. Further, *Plasmodium vivax* invasion was reduced in the low-CR1-expressing cells, with high frequency and strong linkage disequilibrium of the L allele of rs2274567 in *P. vivax*-endemic populations [16, 26].

However, the protective effect of low *CR1* expression was not consistent across studies [13, 18–21, 25, 27, 28]. For example, the L/L genotype of the intronic variant rs11118133 was a risk factor for severe malaria in the Thai population [27]. Another study on the Thai population found that a variant in the *CR1* promoter which is related to high expression levels of the gene was associated with protection against cerebral malaria [19]. Similarly, the L/L genotypes of rs2274567 and rs11118133 were associated with severe malaria in populations inhabiting a non-endemic region of India [18]. No significant association between low *CR1* expression and malaria protection was reported in the Chinese population [29], nor by two independent studies on African groups from Gambia [30, 31].

The population genetic approach allows identifying a protective effect of an allele by detecting of signal of positive selection. Kosoy et al. [16, 26] showed the signal of positive selection on the L allele of the *CR1* gene in Sardinians. The study, however, could not explain the conflicting results on the role of the L allele for malaria infection in different populations.

In our study, using whole-genome sequence data, we reveal the fine-resolution selection history of *CR1* and report novel adaptive variants in this gene that are protective against malaria infection. For the first time, we analyze understudied Asian malaria-endemic population groups and compare the identified selection signals with West Africans (Yoruba). We show the population-specific nature of adaptation against malaria in Asia.

## Materials and methods

### Dataset

In order to detect genetic variants under positive selection in various populations, we used 907 high-coverage whole-genome sequencing datasets which are part of the GenomeAsia 100K Pilot datasets [32]. The genome data are available from the European Genome-phenome Archive (EGA) under accession number EGAS00001002921. We selected populations to include in this study based on the geographical distribution of malaria endemicity (Fig 1). Malaria-endemic and non-endemic countries were identified based on a World Health Organization (WHO) report (World Malaria Report 2018). We classified countries with fewer than

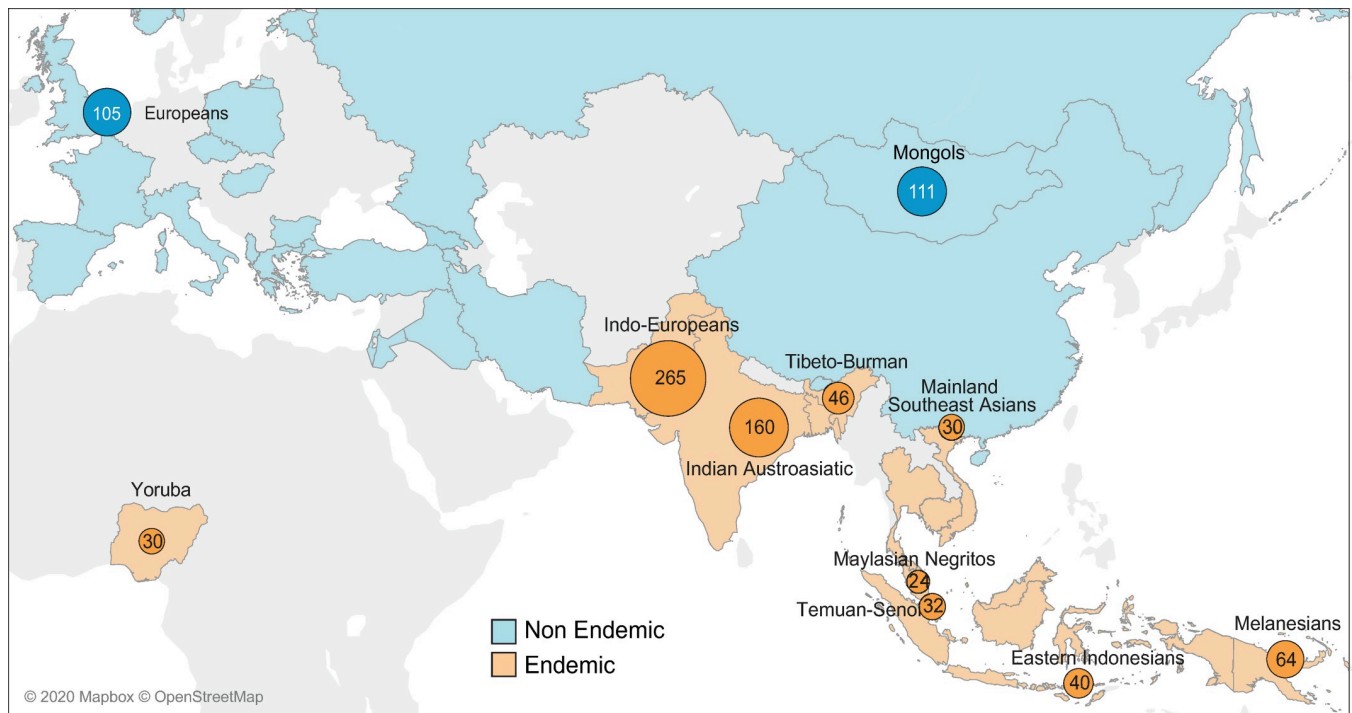

**Fig 1. Geographic locations of the samples included in this study.** The areas colored in blue and orange on the map represent countries where malaria is non-endemic and endemic, respectively, based on a WHO report (World Malaria Report 2018). The 11 population groups analyzed are represented by colored circles. The number in a circle is the number of samples. We used the Tableau v. 2021.2 to create the map images.

100 non-imported malaria cases in 2017 as non-endemic, and countries with at least 100 non-imported malaria cases as endemic (World Malaria Report 2018). Samples that have ambiguous information on endemicity were removed from the dataset. Our dataset includes nine endemic population groups from tropical and subtropical regions of Asia, comprising 46 Tibeto-Burman, 32 Temuan-Senoi, 40 Eastern Indonesians, 30 Mainland Southeast Asians, 265 Indo-Europeans, 24 Malaysian Negritos, 64 Melanesians, 160 Indian Austroasiatic populations, and 30 West Africans (Yoruba). The two non-endemic population groups are 105 Europeans and 111 Mongols (Fig 1). Each population group consists of multiple ethnicities and populations (S1 Table) of similar genetic ancestry. For each population group, we selected a sample size equal to or higher than 24 (Fig 1), which is sufficient to detect selection signals [33–35].

## Identification of positive selection

The genome-wide selection tests using XP-EHH [33], iHS [34], and PBS [35] have been performed in the previous study [36], summarized in DOI: 10.13140/RG.2.2.13261.56804/1. In the current study, we thoroughly examined the outputs of the selection tests for the *CR1* gene and the 50kb upstream and downstream regions (GRCh37, chr1:207,669,473–207,815,110). We excluded the region containing amino acid tandem repeats (chr1:207,697,000–207,738,000) (S1 Fig), which are known to be copy number variations [2, 29, 37]. Due to the complexity of the repeat region, the sequencing quality was insufficient for the identification of SNPs.

A detailed description of the methods of the genome-wide selection tests has been previously reported [36]. The iHS was calculated for each of the 11 population groups (nine

endemic and two non-endemic) independently. For the XP-EHH analysis, each of the nine endemic population groups was compared with one of the non-endemics (Europeans and Mongols) for a total of 18 tests. The PBS test was performed for nine population trios that included one of the endemic population groups and both non-endemics (Europeans and Mongols).

To evaluate the significance of the selection signals in the *CR1* gene, we calculated the percentile ranks of the standardized iHS and XP-EHH scores for each Single Nucleotide Polymorphism (SNP) and calculated the PBS scores for each 10 kb-window genome-wide. We ranked the standardized iHS, XP-EHH, and PBS scores from the largest positive to the smallest negative values and selected the top 5% of the distributions to determine signals of positive selection, as implemented in previous studies [33–35]. For XP-EHH and PBS, we performed a "one-sided test" by taking top 5% of positive values, whereas for iHS we performed a "two-sided test" by taking top 2.5% of positive values and bottom 2.5% of negative values (S2–S6 Tables). The genome-wide percentile ranks were calculated for each population group independently. As the whole-genome distribution of genetic variants for each population group represents neutral variants and is affected by the population history (i.e., ancient migrations and admixture), we defined the variants with higher ranks that are out of the distribution of neutral variants. The variants with the highest ranks are considered to be under selection.

To identify specific mutations favored by selection, we calculated SAFE scores for the *CR1* gene locus using iSAFE tool [38] with the—SAFE option. To determine the ancestral and derived allelic states of the variants in the *CR1* gene locus, we used *Homo sapiens* Ancestral Allele files available from the Ensembl database (GRCh37). SAFE scores range from -1 to 1 and tend to be maximized for the favored mutations in the studied population group.

### Time to the Most Recent Common Ancestor (TMRCA) estimation

We used RELATE [39] to construct haplotype trees of the *CR1* gene region, plus 50Kb upstream and downstream. This method can estimate the TMRCA of each node of the haplotype tree. For this analysis, we included 470 genomes from five population groups with less admixture (West Africans, Europeans, Mongols, Melanesians, and Indian Austroasiatics) since admixture introduce recombinants which can cause inaccurate haplotype topology. To perform the analysis, we used phased sequence data for the entire chromosome 1 [32] along with information on the ancestral type of alleles included in the RELATE package and the genome recombination rate map from the 1000 Genome Project [40]. In order to assess the robustness of the TMRCA estimates, we repeated the entire RELATE analysis 100 times for 100 different sets of 120 randomly sampled genomes, which included 24 samples from three malaria-endemic (Yoruba, Indian Austroasiatic, Melanesian) and two non-endemic (Europeans and Mongols) population groups. The TMRCA was scaled by the parameters of mutation rate ($1.25e^{-8}$/bp/generation), generation time (28 years), and effective population size ($N_e$ = 20,000, 30,000, and 40,000). We iterated the analysis 100 times for each effective population size. We calculated the pairwise linkage disequilibrium (LD) score for SNPs between exon 22 and exon 33 using PLINK 1.9 [41].

## Results

### Positive selection on the CR1 gene in malaria-endemic populations

We assessed selection signals on the *CR1* gene locus in nine malaria-endemic and two non-endemic population groups (Fig 1 and S1 Table, see details in Materials and Methods). The endemic population groups are diverse Asian populations, except one from Africa. The selected non-endemic population groups are Europeans and Mongols.

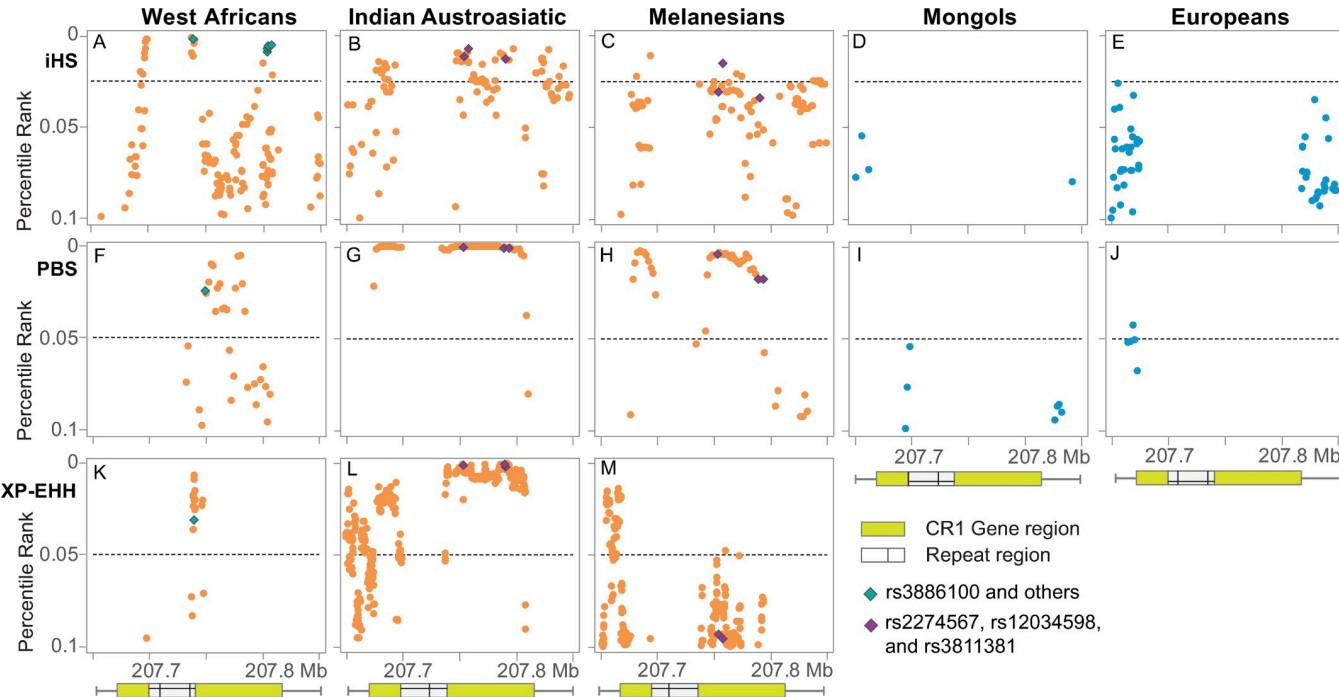

**Fig 2. Genome-wide percentile ranking of three selection tests.** The standardized iHS [34] (A, B, C, D, and E), PBS [35] (F, G, H, I, and J), and XP-EHH [33] (K, L, and M) values across the *CR1* gene region (50kb upstream and downstream) are plotted for the five population groups (three endemic and two non-endemic). The XP-EHH results were the three endemic population groups versus Mongols. The two non-endemic PBS results (I and J) are plotted from the tests for Indian Austroasiatic populations, Mongols, and Europeans. Dots represent SNPs (iHS and XP-EHH) or windows (PBS) having percentile ranking values equal to or lower than top 0.10 (Y axis) over the Mbp position on chromosome 1 (X axis). The *CR1* gene and repeat region on the X axis are indicated as green and mesh bars under the plots, respectively. Across the three methods, here we show SNPs with a percentile ranking equal to or lower than 0.1 for the endemic and non-endemic population groups. In addition, green and purple diamonds indicate the SNPs/windows associated with the *CR1* expression level (rs2274567, rs3811381, rs3886100, rs11803956, rs12041437, rs17186848, and rs11803366) and erythrocyte sedimentation rate (rs12034598).

All three tests show significant selection signals (top 5% of the percentile rank) for the three endemic population groups of Indian Austroasiatic, Melanesian, and West African (Yoruba) (Fig 2, S2–S6 Figs). The signals are particularly strong and robust for Indian Austroasiatic populations, and the adaptive SNPs are located across the entire gene region: 81/657 SNPs identified by iHS, 536/1,436 SNPs by XP-EHH, and 49/109 windows by PBS are detected to occur in the top 5% of the percentile rank (S2–S6 Tables). The selection signals are less significant in Melanesians than in Indian Austroasiatic populations: 24/657 SNPs identified by iHS, 63/1,436 SNPs by XP-EHH, and 37/109 windows by PBS (Fig 2). West Africans had the smallest number of adaptive SNPs: 24/641 SNPs detected by iHS, 23/1436 by XP-EHH, and 16/109 windows by PBS (Fig 2). The selection signals identified in the other endemic population groups show less robust results (S2 Table).

### Favored variants of positive selection

To examine the driving force behind the positive selection on the *CR1* gene, we performed functional annotation of the variants under positive selection in each population group. The two non-synonymous variants rs2274567 and rs3811381, which have been reported to influence the expression level of *CR1* [21, 23–25], have the top 1.12% and 1.28% of the percentile rank in the whole genome iHS tests, respectively, on the Indian Austroasiatic populations. The two variants are the 10th and 13th highest ranks, respectively, out of 657 SNPs located in the *CR1* gene region (Fig 2 and S2 and S3 Figs).

In addition, we detected rs12034598 as a novel variant associated with malaria protection since appeared as the highest percentile ranked SNP in the *CR1* region for the Indian Austro-asiatic populations (intron 24, top 0.71% of the percentile rank), which is also the second-highest (top 1.51% of the percentile rank) for Melanesians (Fig 2). rs12034598 is known to be associated with Erythrocyte Sedimentation Rate (ESR) [42, 43], and the allele under positive selection expresses a slow ESR phenotype. The other top-ranked variants listed in S3–S5 Tables for the Indian Austroasiatic populations in both the iHS and XP-EHH analyses are intronic SNPs, whose functions have not been reported.

The iSAFE [38] results support the three functionally annotated variants as the driving force of this positive selection. In the estimated SAFE scores for the 657 SNPs in the *CR1* gene region, for Indian Austroasiatic populations, rs12034598, rs3811381, and rs2274567 were ranked as the top 4th, 12th, and 16th, respectively, with the high scores ranging from 0.23 to 0.25 (S7 Table). Thus, the three SNPs, rs12034598, rs3811381, and rs2274567, are possible candidates for adaptive variants against malaria infection.

## Adaptive haplotypes and their age

We inferred the history of selection on the low *CR1* expression allele L (G, G, T nucleotides for rs2274567, rs3811381, rs11118133, respectively) in the malaria-endemic population groups by constructing haplotype trees and estimating the Time to the Most Recent Common Ancestor (TMRCA) of the haplotypes using RELATE [39]. RELATE identified 17 haplotype blocks in the region, and trees were constructed for each block from phased haplotype sequences of 120 individuals from three selected endemic (Indian Austroasiatic populations, Melanesians, and West Africans) and two non-endemic population groups (Europeans and Mongols). Admixed populations that possess multiple ancestries were not included in this analysis to avoid recombinant haplotypes.

Across the haplotype blocks, the haplotype trees show similar patterns of phylogeny, especially for the trees of the haplotype blocks containing the regions from exon 22 to exon 33 (Fig 3 and S7 Fig). The haplotypes including exon 22 have two distinct haplogroups, defined by six SNPs that include rs2274567. Since rs2274567 is involved in the CR1 gene expression levels, we designated the two haplogroups as low (L) and high (H) expression level haplogroups (Fig 3). Interestingly, most Indian Austroasiatic and Melanesian haplotypes belong to the L haplogroup, while non-endemic haplotypes are more frequent in the H haplogroup (Fig 3). The intronic variant rs12034598 characterizes the most frequent subclade of the L haplogroup. This sub-haplogroup, defined by rs12034598, is designated as the 'LS haplogroup' because the variant is associated with slow ESR. The LS haplogroup clearly shows characteristics of recent positive selection in the Indian Austroasiatic and Melanesian endemic population groups, including a star-like phylogeny, high frequency, and short branch length.

The TMRCA of the L and LS haplogroups were estimated to be 112–690 thousand years ago (kya) and 48–155 kya, respectively (S8 Fig). The range of the TMRCA is based on 100 bootstrap resampling from the total samples. This estimated TMRCA is in a range similar to the haplotype tree of the region of exon 33. For example, the TMRCA of the haplogroup including the L allele of rs3811381 (exon 33) is 65–323 kya (S9 Fig), which overlaps with the TMRCA of the LS haplogroup. The similar phylogeny and TRMCA between exons 22 and 33 suggest strong LD ($R^2 = 0.89$) throughout the region between the two SNPs (rs2274567 in exon 22, rs3811381 in exon 33), although the haplotype blocks have separated due to the long distance between the regions.

The TMRCA of the H haplogroup was estimated to be older. It shows greater genetic diversity within the cluster than that of the L haplogroup. Endemic West Africans show a higher

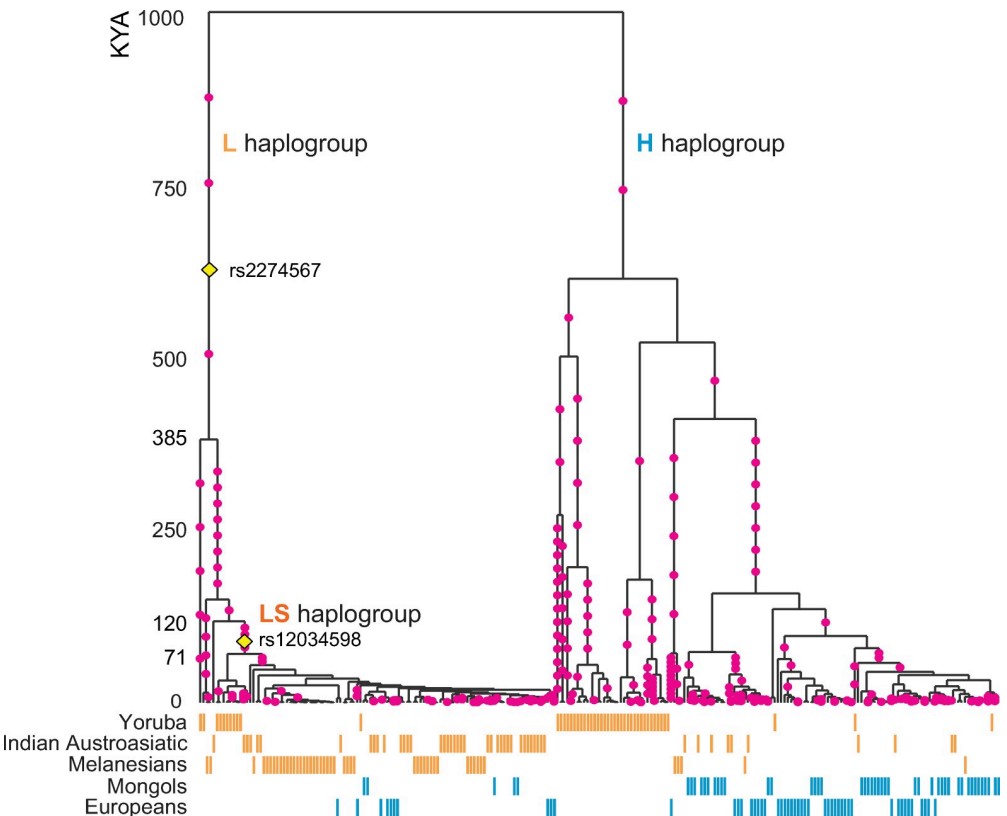

**Fig 3. A coalescent tree of the *CR1* locus.** For the sub-region of the gene locus (intron 20 to intron 27: ~16.7 kb), a coalescent tree was estimated by RELATE [39]. The tree is one estimate out of the 100 replicates, as described in Method. The pink dots on the tree branches represent mutations (SNPs) assigned to the lineages of the tree. Vertical and short bars below the tree correspond to the tips of the trees (each haplotype) of five population groups used to construct the tree. The yellow diamonds indicate the locations of rs2274567 exon 22 SNP and rs12034598 intron 24 SNP on the tree. We detected two distinct haplogroups in the tree, defined by rs2274567, and designated as the L and H haplogroups. The L haplogroup is more frequent in Indian Austroasiatic and Melanesian populations than in Mongols and Europeans. The LS haplogroup is defined by two SNPs, rs2274567 and rs12034598.

haplotype frequency of the H haplogroup (76.7%) than the L haplogroup (23.3%) (Fig 3), which is within a similar range to that of the other African populations (70~80%, the H allele in the AFR populations of 1000 Genomes Project). None of the three favored SNPs of the L allele are detected in the three selection tests (Fig 2), suggesting no significant evidence selection on the L haplogroup in West Africans.

## Population-specific positive selection

The West African population also shows significant selection signals across the three tests. These signals involve different variants than the signals in the Indian Austroasiatic and Melanesian population groups (Fig 2, S2–S6 Figs, and S2 Table). We retrieved information about the expression level of the adaptive variants identified by iHS and XP-EHH from the Genotype-Tissue Expression (GTEx) Data [44] and found that the adaptive alleles of the six SNPs were associated with significantly lower expression levels of *CR1* in brain tissues in mainly Europeans (see the six SNPs in S10 Fig). For example, rs3886100 showed a high percentile rank (top 0.37%) by iHS, corroborated by the XP-EHH (top 3.10%) and PBS (top 2.45%) analyses (Fig 2). The major (G) allele of rs3886100 occurred at a frequency of 97% in West Africans

and was associated with low expression levels of *CR1* in the brain. Therefore, West Africans have obtained adaptive low expression *CR1* variants independently from Indian Austroasiatic and Melanesian population groups.

The different variants under positive selection across population groups suggest population-specific selection histories and convergent evolution of human adaptation to malaria (Fig 3).

## Discussion

We analyzed the signals of positive selection in the *CR1* gene in diverse malaria-endemic populations using genome sequencing datasets. As a result, we identified two novel adaptive variants of the *CR1* gene: rs12034598 and rs3886100. The variant rs12034598 is likely the driving force of the recent positive selection on the L allele of the *CR1* in Indian Austroasiatic and Melanesian populations. The G allele of rs12034598, which is associated with low ESR and reduced inflammation [42, 43], could be advantageous in reducing the risk of severe malaria infection. This protective effect becomes stronger in combination with the low expression of *CR1*, determined by rs2274567 and rs3811381.

Our estimates show that rs12034598 occurred before the out-of-Africa migration, 50–160 kya (Fig 3), thus both Africans and non-Africans carry the mutation. Only after the migration, recent malaria breakouts in Asia triggered positive selection on the LS haplogroup independently in the Indian Austroasiatic and Melanesian populations. This is supported by the separate clustering of the LS haplogroup by the populations in the haplotype tree (Fig 3).

We demonstrated the selection pressure on the LS haplogroup by comparing transmission rates with mortality from malaria infection. The frequency of the LS haplogroup (including the G allele of rs2274567 and the G allele of rs12034598) in each ethnic group and the mortality rates from malaria infection are shown in the maps of India and Island Southeast Asia (Fig 4, S8 Table, S11 and S12 Figs). The high frequency of the LS haplogroup is observed in the endemic groups with relatively higher mortality rate caused by *P. falciparum*: Melanesian ethnic groups living on the islands of Papua New Guinea and New Britain (LS frequency range: 75% to 100%, malaria mortality: 9.6 per 100,000 people in 2017) [45], Indian Austroasiatic ethnic groups living in East India (LS frequency range: 64% to 100%, malaria mortality: 1.6~72 per 100,000 people in 2017) [45], and the Tibeto-Burman ethnic groups living in Northeast India (LS frequency range: 55% to 69%, malaria mortality: 2.8~16.6 per 100,000 people in 2017) [45] (S8 Table, Fig 4). Although we obtained the transmission and mortality rate data only from recent records [45], the East India and Papua New Guinea regions are known to have been endemic for a long time.

The co-occurrence of the LS haplogroup and malaria endemicity is notable in the ethnic groups within India, supporting the hypothesis of ongoing positive selection on the LS haplogroup. In East India, 9,348,000 confirmed malaria cases and 16,310 malaria-related deaths were reported in 2017. The cases were concentrated in three states: Odisha, Chhattisgarh, and Jharkhand (S13 Fig). In these hilly and forested areas, the hot and humid climate promotes mosquito breeding. Many of these malaria high-transmission zones are inhabited by tribal populations [46], which are difficult to access for malaria control due to a lack of infrastructure. Thus, malaria occurrence in these areas has been pervasive and sustained, and *P. falciparum* and *P. vivax* parasites can be found there in equal proportions [46]. CR1 expressed on the RBC could modulate invasion by *P. vivax* as well as by *P. falciparum* [16, 47]. A meta-analysis performed in East India showed that low expression of CR1 on RBCs was protective against cerebral malaria [13], which is known to be caused by *P. falciparum*, supporting our finding of positive selection on the LS haplogroup.

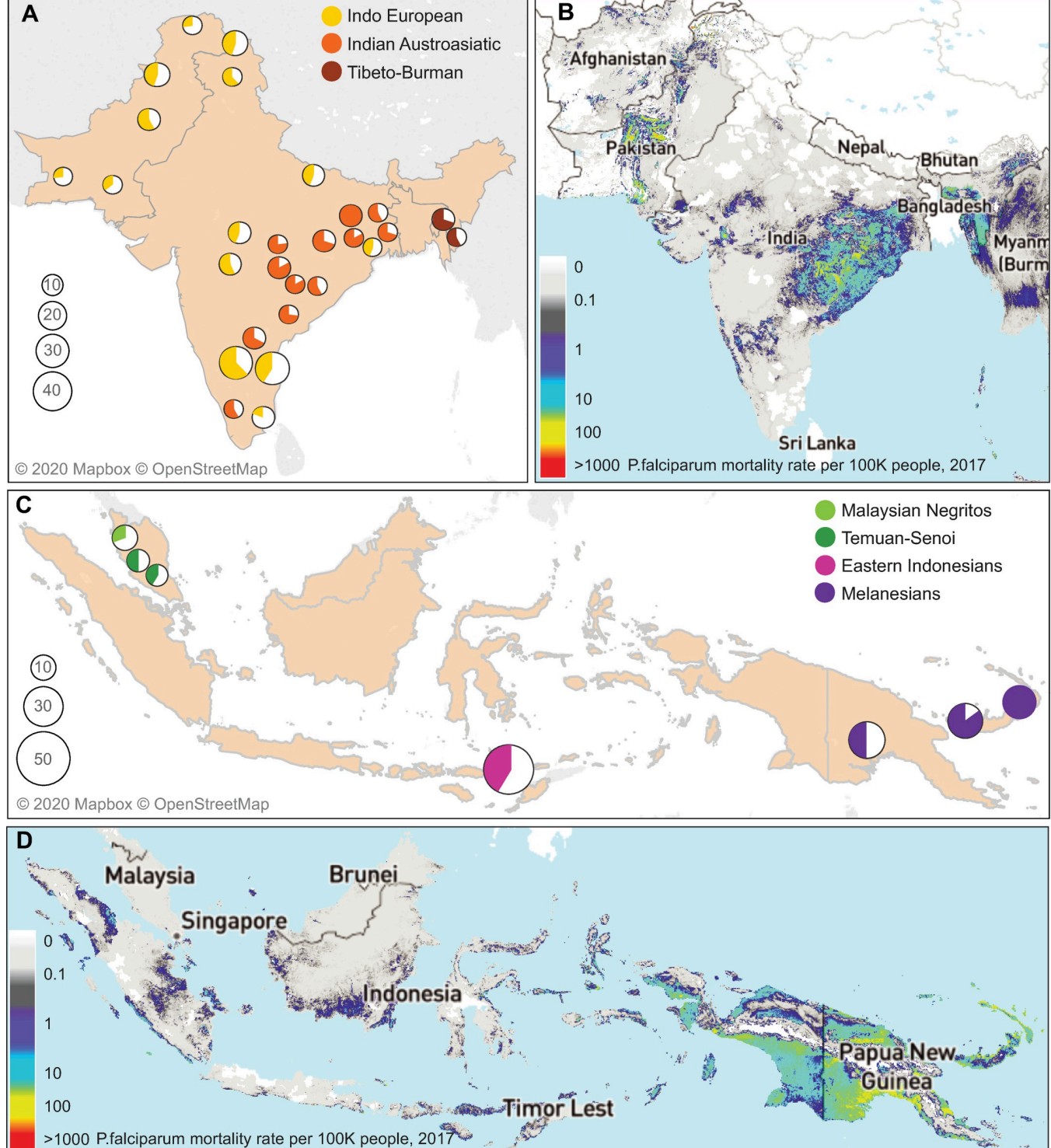

**Fig 4. Frequency distribution of the LS haplogroup and the malaria mortality rate.** For the ethnic groups living in South and Island Southeast Asia, the haplotype frequency of the LS haplogroup (A and C) and the malaria mortality rate (B and D) are shown on the geographic map. The pie charts show the frequency of the LS haplogroup in each ethnic group with a sample size equal to or greater than 10. The colors in the pie charts represent the population group which the ethnic group belongs to. The mortality rate per 100,000 people is shown as colors in the map, which was retrieved from the malaria atlas project [45]. The haplotype frequencies for all ethnic groups are provided in the S8 Table.

For the Malaysian indigenous people (Temuan and Senoi), the iHS analysis showed a significant selection signal at the *CR1* promoter (S2 Fig). One of the SNPs in the promoter, rs9429942 (T allele), was reported to be associated with high expression of erythrocyte CR1 as well as protection against cerebral malaria in Thai populations [19]. The potential selection pressure on the T allele of rs9429942 (top 1.5%) is supported only by iHS. The high CR1 expression in erythrocytes may contribute to a high clearance of immune complexes [1, 2], and decrease inflammation caused by malaria [15, 42, 43]. Despite the recent decrease in malaria parasite numbers and mortality rates in Malaysia (Fig 4 and S11 and S12 Figs), the habitat in the tropical forests where the Temuan and Senoi people live might maintain the selection pressure on the *CR1* gene variant [48].

We did not detect any significant selection signal in the Dai group, who are recent migrants from Thailand to Southern China. Over many generations, changes in malaria environments may have relaxed the selection pressure on the *CR1* gene.

The positive selection on the low expression of *CR1* in West Africans is detected on a different variant, rs3886100. This population-specific manner of positive selection on the low expression of the *CR1* suggests convergent evolution of human adaptation to malaria, probably due to the different malaria environments and endemicity across different regions (S11 and S12 Figs). For example, a previous study identified a higher gene frequency for the low expression L allele on a specific polymorphism in malaria-endemic regions in Asia but not in Africa [16, 49]. In our study, we found positive selection operating on the low expression alleles in both Asians and Africans, but on different variants, indicating a population-specific manner of positive selection. Thus, our results explain the contradictory results of the previous studies.

## Conclusions

Our results clarify discrepancies reported in previous studies on the role of CR1 on the pathogenesis of malaria. In particular, we show strong evidence of positive selection on the CR1 low expression variants in specific malaria endemic regions. We report two new adaptive variants, rs12034598 in Indian Austroasiatic and Melanesian populations and rs3886100 in West Africans. This research highlights the advantages of a population genetic approach, which utilizes whole genome datasets of diverse ethnic groups to identify novel variants that are protective against the severe form of malaria. This approach does not require patient data or individual clinical records and can be applied to other endemic infectious diseases (e.g., leishmaniasis, dengue, yellow fever, leprosy). However, future studies will be needed to reveal the difference in the expression level of the selected haplotype to develop personalized medicine against severe malaria.

## Supporting information

**S1 Fig. CR1 protein structure.** Graphical annotation of the structure of the CR1 locus at different levels. Location of the CR1 gene on chromosome 1 is indicated by a red vertical line (UCSC genome browser top image). The genome annotation represents the structure of the two major CR1 isoforms H and L with the introns as blue arrows and exons as vertical blue lines. Black arrows indicate the locations on the CR1 gene of the four SNPs which can determine expression levels (rs73689510 on exon 19, rs2274567 on exon 22, rs11118133 on intron 27 and rs3811381 on exon 33) and of a SNP which can affect ESR (rs12034598 on intron 24). The locations and orientations of the Low-Copy Repeats (LCRs) are represented as horizontal arrows below the genome annotation while absence of a particular LCR is indicated as a deletion (white rectangular box). The transcript annotation represents the number of exons (boxes) in each isoform. Each LCR consists of 8 exons and their locations on the genome is

highlighted by shaded areas. The protein annotation shows the organization of the long homologous repeat regions (LHRs). Cylinders of different colours represent different LHRs. The white box labelled TM denotes the region encoding the transmembrane domain. In addition, the protein level depicts the CR1 functional domains with each circle representing a separate short consensus repeat (SCR). Each LHR is in fact composed by 7 different SCRs. The longer isoform H possess an additional LHR S not found in the shorter isoform which can increase the number of binding sites to the complement proteins (white spheres and ovals). Darker coloured spheres at the protein level represent SCRs involved in the binding to the complement and to malaria parasite proteins (grey and black ovals). White empty spheres indicate the locations of the Knops blood group antigen erythrocyte polymorphisms.
(PDF)

**S2 Fig. Genome-wide percentile ranking of the standardised iHS negative values.** The top 10% most negative values are plotted in the CR1 gene region including 50kb upstream and downstream for each of the 11 population groups analysed. Dots and triangles represent SNPs having percentile ranking values equal or lower then 0.10 ($< 10\%$) indicated on the Y axis over the location on chromosome 1 (X axis) in Mega bases (Mb). The green bar under the X axis represents the CR1 gene region, and the mesh area indicates repeats. The regions 50kb upstream and downstream of the CR1 gene are indicated as a line. In the DNA repeat region, no SNPs were called. In addition, purple triangles indicate the locations of rs2274567 exon 22, rs12034598 intron 24 and rs3811381 exon 33 SNPs.
(PDF)

**S3 Fig. Genome-wide percentile ranking of the standardised iHS positive values.** The top 10% most positive are plotted in the CR1 gene region including 50kb upstream and downstream for each of the 11 population groups analysed. Dots and triangles represent SNPs having percentile ranking values equal or lower then 0.10 ($< 10\%$) indicated on the Y axis over the location on chromosome 1 (X axis) in Mega bases (Mb). The green bar under the X axis represents the CR1 gene region, and the mesh area indicates repeats. The regions 50kb upstream and downstream of the CR1 gene are indicated as a line. In the DNA repeat region, no SNPs were called. In addition, dark green triangles indicate the locations of SNPs showing significantly low expression levels of CR1 in brain tissues (S10 Fig): rs3886100, rs11803956, rs12041437, rs17186848, rs12034383, and rs11803366.
(PDF)

**S4 Fig. Genome-wide percentile ranking of the standardised XP-EHH tests against Mongol.** The XP-EHH values are plotted in the CR1 gene region including 50kb upstream and downstream for each of the 9 endemic population groups compared to the reference population of Mongols (non-endemic). Dots and triangles represent SNPs having percentile ranking values equal or lower then 0.10 (top 10%) indicated on the Y axis over the location of SNPs on chromosome 1 (X axis) in Mega bases (Mb). The green bar under the X axis represents the CR1 gene region, and the mesh area indicates repeats. The regions 50kb upstream and downstream of the CR1 gene are indicated as a line. In the DNA repeat region no SNPs called. We detected a larger number of SNPs having a percentile ranking of top 5% in the two endemic population groups (Indian Austroasiatic, Melanesians, and West Africans) in the CR1 gene region. Purple triangles indicate rs2274567, rs12034598, and rs3811381.
(PDF)

**S5 Fig. Genome-wide percentile ranking of the standardised XP-EHH tests against Europeans.** The XP-EHH values are plotted in the CR1 gene region including 50kb upstream and downstream for each of the 9 endemic population groups compared to the reference

population of Europeans (non-endemic). Dots and triangles represent SNPs having percentile ranking values equal or lower then 0.10 (top 10%) indicated on the Y axis over the location of SNPs on chromosome 1 (X axis) in Mega bases (Mb). The green bar under the X axis represents the CR1 gene region, and the mesh area indicates repeats. The regions 50kb upstream and downstream of the CR1 gene are indicated as a line. In the DNA repeat region no SNPs called. We detected a larger number of SNPs having a percentile ranking of top 5% in the two endemic population groups (Indian Austroasiatic, Melanesians, and West Africans) in the CR1 gene region. Purple triangles indicate rs2274567, rs12034598, and rs3811381.
(PDF)

**S6 Fig. Genome-wide percentile ranking of the PBS results.** The branch length of endemic population group are plotted in the CR1 gene region including 50kb upstream and downstream for each of the endemic population groups versus two non-endemic population groups, Mongols and Europeans. Dots and triangles represent SNPs having percentile ranking values equal or lower then 0.10 (top 10%) of midpoints of windows on chromosome 1 in Mega bases (Mb) on the X axis. The green bar under the X axis represents the CR1 gene region, and the mesh area indicates repeats. The regions 50kb upstream and downstream of the CR1 gene are indicated as a line. In the DNA repeat region no SNPs called. Purple triangles indicate the locations of windows containing rs2274567 exon 22, rs12034598 intron 24, and rs3811381 exon 33 SNPs.
(PDF)

**S7 Fig. Coalescence trees of the CR1 gene region estimated by RELATE.** The chromosome position of the region for each tree is shown on the top of the tree. The region for the four tree encompasses from intron 27 to intron 35. The tree in panel C includes exon 33 (rs3811381). Red dots represent mutations (SNPs) assigned to a branch where the ancestral and derived allele were not flipped. The estimated coalescence time is shown on the Y axis in years. Vertical colored lines below the tree represent individuals in the five population groups.
(PDF)

**S8 Fig. Dumbbell plot of the estimation of the Time to most recent common ancestor (TMRCA).** The estimations for the L haplogroup (A) and LS haplogroup (B) from 100 replications are plotted. The coalescence tree of Fig 3 in the main text is one of the results of this replication. Blue horizontal lines represent the time for beginning (left) and the end (right) of the branch to which the two SNPs were assigned. A red dot on the blue line indicates the middle point of the branch. The vertical black line indicates the mean of the middle points. The time was scaled by three different effective population sizes (Ne = 20000, 30000, and 40000), and a generation time of 28 years was used.
(PDF)

**S9 Fig. Dumbbell plot of the estimation of the Time to most recent common ancestor (TMRCA) of the tree of S7C Fig.** We estimated the coalescence time for the two branches which include rs3811381 (A) and rs12734030 (B), respect, and perform the analysis 100 times. Blue horizontal lines represent the time for beginning (left) and the end (right) of each of the branches. A red dot on the blue line indicates the middle point of the branch. The vertical black line indicates the mean of the middle points. The time was scaled by three different effective population sizes (Ne = 20000, 30000, and 40000), and a generation time of 28 years was used.
(PDF)

**S10 Fig. CR1 expression levels in Brain tissue.** The CR1 gene expression levels in brain tissue of the six SNPs under positive selection are shown in the violin plots. Allele-specific cis-eQTLs

in human brain hippocampus tissue are retrieved from the Genotype-Tissue Expression (GTEx Analysis Release V8 dbGaP Accession phs000424.v8.p2) database. The teal region indicates the density distribution of the samples in each genotype. The white line in the box plot (black) shows the median value of the expression of each genotype. All SNPs show significant difference (P value under the SNP rs ID) of expression level between genotypes.
(PDF)

**S11 Fig. The maps of the *Plasmodium falciparum* parasite rate in 2000~2019.** The maps were retrieved from the malaria atlas project [45].
(MOV)

**S12 Fig. The maps of predicted all-age *Plasmodium falciparum* mortality rate in 2000~2019.** The maps were retrieved from the malaria atlas project [45].
(MOV)

**S13 Fig. Transmission map of malaria in India.** Background map indicates the malaria transmission rate in each state of the country based on the Annual Parasite Incidence (API) which denotes malaria cases per 1000 population amongst individuals of any age. API values were calculated based on the statistics of malaria cases obtained from National Vector Borne Diseases Control Programme, India.
(PDF)

**S1 Table. Population compositions and sample size of the 11 population groups included in the study.**
(XLSX)

**S2 Table. The number and proportion of SNPs under positive selection for each population group and test.**
(XLSX)

**S3 Table. The results of iHS tests for each of the 11 population groups.** The significant SNPs are highlighted by colored shadow.
(XLSX)

**S4 Table. The results of XP-EHH vs. Mongols tests for each of the 11 population groups.** The significant SNPs are highlighted by colored shadow.
(XLSX)

**S5 Table. The results of XP-EHH vs. Europeans tests for each of the 11 population groups.** The significant SNPs are highlighted by colored shadow.
(XLSX)

**S6 Table. The results of PBS tests for each of the 11 population groups.** Each row is a result of a 10-kb window, and the midpoint of the window is indicated in the table. The significant SNPs are highlighted by colored shadow.
(XLSX)

**S7 Table. SAFE scores of the SNPs in the CR1 gene region for each of the 11 population groups.**
(XLSX)

**S8 Table. The frequency of the haplogroups including the two marker SNPs (rs2274567:A/ G and rs12034598:A/G), defining the LS haplogroup for each population.** The possible haplotypes are AA, AG, GA, and GG, and the frequency for each haplotype in each population is

shown in the table.
(XLSX)

## Acknowledgments

We thank Pavel Adamek and Sam Spence for their critical reading and helpful comments on the manuscript. The computational work for this article was partially performed on resources of the National Supercomputing Center, Singapore (https://www.nscc.sg).

## Author Contributions

**Conceptualization:** Elena S. Gusareva, Hie Lim Kim.

**Data curation:** Paolo Alberto Lorenzini.

**Formal analysis:** Paolo Alberto Lorenzini, Elena S. Gusareva, Nurul Adilah Binte Ramli.

**Funding acquisition:** Peter Rainer Preiser, Hie Lim Kim.

**Investigation:** Paolo Alberto Lorenzini, Hie Lim Kim.

**Methodology:** Paolo Alberto Lorenzini, Amit Gourav Ghosh, Nurul Adilah Binte Ramli.

**Project administration:** Hie Lim Kim.

**Resources:** Hie Lim Kim.

**Supervision:** Peter Rainer Preiser, Hie Lim Kim.

**Validation:** Elena S. Gusareva, Amit Gourav Ghosh, Nurul Adilah Binte Ramli, Hie Lim Kim.

**Visualization:** Paolo Alberto Lorenzini, Hie Lim Kim.

**Writing – original draft:** Paolo Alberto Lorenzini.

**Writing – review & editing:** Elena S. Gusareva, Amit Gourav Ghosh, Nurul Adilah Binte Ramli, Peter Rainer Preiser, Hie Lim Kim.

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
