## [Decision Letter · Decision Letter 0]

31 Oct 2022

PONE-D-22-21304Population-specific positive selection on low CR1 expression in malaria-endemic regionsPLOS ONE

Dear Dr. Gusareva,

Thank you for submitting your manuscript to PLOS ONE. After careful consideration, we feel that it has merit but does not fully meet PLOS ONE’s publication criteria as it currently stands. Therefore, we invite you to submit a revised version of the manuscript that addresses the points raised during the review process.

ACADEMIC EDITOR:The reviewers have raised some concerns about the selection signals identified against malaria, and authors are suggested to clarify these concerns in detail.

We look forward to receiving your revised manuscript.

Kind regards,

Hoh Boon-Peng, PhD

Academic Editor

PLOS ONE

Journal Requirements:

“This research was supported by the Singapore Ministry of Education, Academic Research Fund Tier 1 (grant number 2017-T1-001-046). The computational work for this article was performed in part on resources of the National Supercomputing Centre, Singapore (https://www.nscc.sg) supported by Project 12000454.”

“The authors declare no conflict of interest.”

5. We note that Figures 1 and 4 in your submission contain [map/satellite] images which may be copyrighted. All PLOS content is published under the Creative Commons Attribution License (CC BY 4.0), which means that the manuscript, images, and Supporting Information files will be freely available online, and any third party is permitted to access, download, copy, distribute, and use these materials in any way, even commercially, with proper attribution. For these reasons, we cannot publish previously copyrighted maps or satellite images created using proprietary data, such as Google software (Google Maps, Street View, and Earth). For more information, see our copyright guidelines: http://journals.plos.org/plosone/s/licenses-and-copyright.

   a. You may seek permission from the original copyright holder of Figures 1 and 4 to publish the content specifically under the CC BY 4.0 license. 

Reviewers' comments:

Reviewer's Responses to Questions

**Comments to the Author**

1. Is the manuscript technically sound, and do the data support the conclusions?

Reviewer #1: Yes

Reviewer #2: Partly

2. Has the statistical analysis been performed appropriately and rigorously? 

Reviewer #1: Yes

Reviewer #2: Yes

3. Have the authors made all data underlying the findings in their manuscript fully available?

Reviewer #1: Yes

Reviewer #2: Yes

4. Is the manuscript presented in an intelligible fashion and written in standard English?

Reviewer #1: Yes

Reviewer #2: Yes

5. Review Comments to the Author

Reviewer #1: The novelty of this manuscript is two fold as mentioned below

One is finding of NEW SNPs related to protection of malaria and the other, more important than the first one, is finding a population-specific manner of positive selection.

Similar result for the latter case is reported by an example of LCT persistence, which shows different SNPs are target of this selection between African and European. However, such a case of population specific selection is not commonly observed.

However, in this manuscript, there are several unclear points commented as follows.

I will recommend revision of this manuscript for its publication.

Major comments:

1) This manuscript described four SNPS, rs2274567, rs3811381, rs12034598, and rs3886100. But they did not mention the latter two SNPs are novel ones to protect people from malaria, when they appeared at the first time. Later, on line 221 they mentioned “two novel variants” are rs12034598 and rs3886100. If the authors wanted to emphasize this result, description of these two SNPs would be separated the previously known ones and the authors are recommended to make a new paragraph to describe this. The current description of theses alleles may lead confusion to readers.

2) TMRCA estimation (Adaptive haplotypes and their age): RELATE identified 17 haplotype blocks in the region. On line 180, “the haplotype trees show similar pattern of phylogeny, especially for the trees of the haplotype blocks containing the regions from exon2 to exon 33”. When I looked at the S7 Fig., based on the legend the trees are from intron 27 to exon 35. There is no clear correspondence of each tree with exons/introns. On line 196, “suggest LD throughout the region between the two exons” is not appropriately expressed. This should be “suggest strong LD (r2=0.89) throughout the region between the two SNPs (rs2274567 in exon 22, rs3811381 in exon 33)”.

Minor comments:

1) On page6 line 109-110: top-ranked values were selected for iHS: Significant signals for iHS appears either positive and negative. In this case the test should be both sided. If so, the top-ranked values should be 2.5% for each side. But in the present description, we can read 5% for each side because of the sentence of line 107-108. You should distinguish “one-side test” or “ both-side test”.

2) On page 7 line 124: to avoid inaccurate inference. The word of “inaccurate” is vague. Please rewrite this phrase by clear expression.

3) On page 7 line 128-129: “for 100 different individual sets of 120 randomly sampled genomes”. This sentence may be confusing for readers. What do the authors mean “100 different individual sets”? I guess that “100 (different) sets of 120 randomly sampled genomes” is clearer.

4) On page 8 line 155: “have the top 1.12% and 1.28% of the percentile rank in the whole genome”. I am not sure about “percentile rank of what” and “whose whole genome”.

5) On page 9 line 182-183: “defined by six SNPs that include rs2274567. Thus, we designated the two haplogroups as low(L) and high(H) expression level haplogroups”. If the authors use “Thus”, the authors should mention that rs2274567 is involved in the expression level of CR1.

6) On page 10 line 186: "the most frequent subnode". For me this expression reads somewhat strange. This phrase may be “the most frequent subclade”.

7) On page 10 line 192-193: "based on 100 repeated estimates of random sampling". First, Di this mean that this estimation based on 100 bootstrap resampling? Second, resampling is from the total samples or from only L or LS haplogroups.

8) On page 10 line 204: Do the authors mean “minimum selection constraints” as “relaxation of functional constraints” or “minimum selection coefficients”?

9) On page 12 line 238,239,241: The frequency of LS 75% to 100% does not seem to agree to 9.6 per 100,000 people (0.0096%). What is the base of this LS frequency? The same argument is for line 239 and 241.

Reviewer #2: Despite CR1 gene has been extensively studied to be in association with rosseting of red blood cells that subsequently cause microvascular obstruction and eventual severe/cerebral malaria, the complete spectrum of variations across the whole gene and its flanking regulatory regions are remained unknown. This work harnesses the power of next generation sequencing and has sequenced the complete CR1 gene and identify all variations from individuals residing in malaria-endemic and non-endemic regions. Further analyses of positive selection identified SNPs that could act as protective biomarkers against malaria infection / development toward severe/cerebral malaria.

A number of comments on this manuscript are as follow:

1. Previous studies always select a few SNPs of CR1 gene for testing its association to malaria infection. You pointed out that the association could be contrasting. Thus, this should be overcome. In addition, as a gene could have a few hundred variations across human populations, the power of sequencing and subsequent analyses should be harnessed to select candidate SNPs with strong protective effects. Please have an in-depth literature review over this and articulate the specific objective you wish to work out.

2. Do you specify that the force of selection is the lethal cerebral malaria or severe types of malaria (in exception to cerebral one). Some species might be more pathogenic and lethal e.g. P. falciparum infection commonly lead to cerebral malaria, attributed to different pathogenesis pathways. And the distribution of the different species could be different, e.g. P. knowlesi in ISEA. In addition, using malaria case numbers as a measure of endemicity and force of selection is an appropriate proxy, but unfortunately these case numbers do not specify which species of Plasmodium that caused which severe types of malaria. Judging on that, these could serve to drive the selection force differently. Looking at the high number of cases in the current days, I wonder has this force of selection yet been fixed, and thus the method of analysis should be revised? Please clarify these inter-related problems in detail.

3. Since you inferred that the positive selection already occurred before human migrating out of Africa, I suppose that majority of the extant human racial groups should carry the same polymorphic SNPs which are almost fix, in both endemic and non-endemic extant populations. But your data did not find so. What could be the reason?

4. Some samples are carrying more variations in the gene. How do you test and remove the mentioned ‘less admixed’ individuals? How do you affirm that the positively selected variants are not due to ancestry / anthropology, instead of natural selection?

5. Based on the GTEX paper published in 2020, a great majority of 85% of the dataset are of European American. Since European countries are not in the Malaria endemic region, and you also found that there is no signal of selection on CR1 gene among the European samples, you thereby used this public gene expression data (low CR1 expression) as your strong support that infers that this gene is also expressed low among the other multi-racial / ancestries individuals in Asia & Africa. Unless it is tested empirically by any measure of gene expression using RNA from your studies samples, this inference is invalid.

6. The outcome of this work is derived from bioinformatic analyses of individuals with unknown history of malaria infection, and thus the obtained genotypic and allelic frequencies should represent the general polymorphisms in each regional population. However, the conclusion can only be made after a well-designed case-control association test is conducted. As such, your previous work (Gusareva et al., 2021) should have already identified similar results as this current manuscript. I wonder why CR1 analysis was split out from the previous work?

7. The conclusion states that the current findings could be helpful in precision medicine of malaria medical management. Unfortunately, this bioinformatic findings have yet to be extensively tested but the authors already gave strong conclusive statements. I find this misleading and should be removed.

6. PLOS authors have the option to publish the peer review history of their article (what does this mean?). If published, this will include your full peer review and any attached files.

Reviewer #1: No

Reviewer #2: No

---

## [Author Response · Author response to Decision Letter 0]

15 Nov 2022

Reviewer #1: 

The novelty of this manuscript is two fold as mentioned below

One is finding of NEW SNPs related to protection of malaria and the other, more important than the first one, is finding a population-specific manner of positive selection.

Similar result for the latter case is reported by an example of LCT persistence, which shows different SNPs are target of this selection between African and European. However, such a case of population specific selection is not commonly observed.

However, in this manuscript, there are several unclear points commented as follows.

I will recommend revision of this manuscript for its publication.

Major comments:

1) This manuscript described four SNPS, rs2274567, rs3811381, rs12034598, and rs3886100. But they did not mention the latter two SNPs are novel ones to protect people from malaria, when they appeared at the first time. Later, on line 221 they mentioned “two novel variants” are rs12034598 and rs3886100. If the authors wanted to emphasize this result, description of these two SNPs would be separated the previously known ones and the authors are recommended to make a new paragraph to describe this. The current description of theses alleles may lead confusion to readers.

Response: We amended the main text in several parts in order to describe the two SNPs as novel variants associated with malaria. The Conclusions section was also amended to emphasize the two new adaptive variants, rs12034598 in Indian Austroasiatic and Melanesian populations and rs3886100 in West Africans, associated with low levels of CR1 expression (P. 16. L. 326-329).

2) TMRCA estimation (Adaptive haplotypes and their age): RELATE identified 17 haplotype blocks in the region. On line 180, “the haplotype trees show similar pattern of phylogeny, especially for the trees of the haplotype blocks containing the regions from exon2 to exon 33”. When I looked at the S7 Fig., based on the legend the trees are from intron 27 to exon 35. There is no clear correspondence of each tree with exons/introns. On line 196, “suggest LD throughout the region between the two exons” is not appropriately expressed. This should be “suggest strong LD (r2=0.89) throughout the region between the two SNPs (rs2274567 in exon 22, rs3811381 in exon 33)”.

Response: The line 180 (now P.10, L. 206-209) states that the region is from exon 22 (not exon2) to exon 33: “Across the haplotype blocks, the haplotype trees show similar patterns of phylogeny, especially for the trees of the haplotype blocks containing the regions from exon 22 to exon 33 (Fig 3 and S7 Fig).” Figure 3 includes exon 22 whereas trees in the supplementary figures include downstream regions as well.

We amended the line 196 (now P. 11, L. 234-235) as suggested.

Minor comments:

1) On page6 line 109-110: top-ranked values were selected for iHS: Significant signals for iHS appears either positive and negative. In this case the test should be both sided. If so, the top-ranked values should be 2.5% for each side. But in the present description, we can read 5% for each side because of the sentence of line 107-108. You should distinguish “one-side test” or “ both-side test”.

Response: We amended the Methods section accordingly to clarify rankings for the XP-EHH, iHS, and PBS tested performed in our study (P. 6, L. 118-120). Particularly, “For XP-EHH and PBS, we performed a “one-sided test” by taking top 5% of positive values, whereas for iHS we performed a “two-sided test” by taking top 2.5% of positive values and bottom 2.5% of negative values”.

2) On page 7 line 124: to avoid inaccurate inference. The word of “inaccurate” is vague. Please rewrite this phrase by clear expression.

Response: We amended this sentence on P. 7, L. 134-135 and provided more information. 

3) On page 7 line 128-129: “for 100 different individual sets of 120 randomly sampled genomes”. This sentence may be confusing for readers. What do the authors mean “100 different individual sets”? I guess that “100 (different) sets of 120 randomly sampled genomes” is clearer.

Response: We modified the sentence on P. 7, L. 139 as suggested.

4) On page 8 line 155: “have the top 1.12% and 1.28% of the percentile rank in the whole genome”. I am not sure about “percentile rank of what” and “whose whole genome”.

Response: We amended the text on P. 9, L. 180-181 to clearly specify the percentile ranks for the two adaptive variants, rs2274567 and rs3811381, in the iHS tests on the Indian Austroasiatic of which method and in which populations.

5) On page 9 line 182-183: “defined by six SNPs that include rs2274567. Thus, we designated the two haplogroups as low(L) and high(H) expression level haplogroups”. If the authors use “Thus”, the authors should mention that rs2274567 is involved in the expression level of CR1.

Response: We amended the text as suggested (P. 10, L. 209).

6) On page 10 line 186: "the most frequent subnode". For me this expression reads somewhat strange. This phrase may be “the most frequent subclade”.

Response: We amended the text as suggested (P. 10, L. 213).

7) On page 10 line 192-193: "based on 100 repeated estimates of random sampling". First, Di this mean that this estimation based on 100 bootstrap resampling? Second, resampling is from the total samples or from only L or LS haplogroups.

Response: We meant 100 bootstrap resampling of the total samples. We amended the text to make it clear (P. 11, L. 230-231).

8) On page 10 line 204: Do the authors mean “minimum selection constraints” as “relaxation of functional constraints” or “minimum selection coefficients”?

Response: We mean no significant evidence of selection. We modified main text to make it clearer (P. 12, L. 242).

9) On page 12 line 238,239,241: The frequency of LS 75% to 100% does not seem to agree to 9.6 per 100,000 people (0.0096%). What is the base of this LS frequency? The same argument is for line 239 and 241.

Response: As first value inside the brackets, we meant the LS frequency, whereas as a second value, we indicated malaria mortality rate. We amended the main text accordingly to distinguish the two types of information (P. 13, L. 276-279).

Reviewer #2: 

Despite CR1 gene has been extensively studied to be in association with rosseting of red blood cells that subsequently cause microvascular obstruction and eventual severe/cerebral malaria, the complete spectrum of variations across the whole gene and its flanking regulatory regions are remained unknown. This work harnesses the power of next generation sequencing and has sequenced the complete CR1 gene and identify all variations from individuals residing in malaria-endemic and non-endemic regions. Further analyses of positive selection identified SNPs that could act as protective biomarkers against malaria infection / development toward severe/cerebral malaria.

A number of comments on this manuscript are as follow:

1. Previous studies always select a few SNPs of CR1 gene for testing its association to malaria infection. You pointed out that the association could be contrasting. Thus, this should be overcome. In addition, as a gene could have a few hundred variations across human populations, the power of sequencing and subsequent analyses should be harnessed to select candidate SNPs with strong protective effects. Please have an in-depth literature review over this and articulate the specific objective you wish to work out.

Response: As written in the introduction (P. 3-4, L. 48-64), we made a very in-depth literature research which describes in detail the associations of the L and H alleles of the CR1 gene with the protection or susceptibility against malaria in several human populations. This literature research includes 15 citations (refs. 13-28). From this research we have observed that sometime the high expression of the CR1 gene (H allele) was reported to be associated with malaria protection, despite the low expression (L allele) should be the one conferring protection. Our objective was to clarify this discrepancy and report the associations in the same ethnic groups by utilizing whole genome sequencing data and methods of population genetics which should provide stronger evidence of whether the low expression or the high expression is the one that is conferring protection. We have described in detail results of this objective in our results section.

2. Do you specify that the force of selection is the lethal cerebral malaria or severe types of malaria (in exception to cerebral one). Some species might be more pathogenic and lethal e.g. P. falciparum infection commonly lead to cerebral malaria, attributed to different pathogenesis pathways. And the distribution of the different species could be different, e.g. P. knowlesi in ISEA. In addition, using malaria case numbers as a measure of endemicity and force of selection is an appropriate proxy, but unfortunately these case numbers do not specify which species of Plasmodium that caused which severe types of malaria. Judging on that, these could serve to drive the selection force differently. Looking at the high number of cases in the current days, I wonder has this force of selection yet been fixed, and thus the method of analysis should be revised? Please clarify these inter-related problems in detail. 

Response: We thank you the reviewer to highlight this very important point on the different pathogenesis and prevalence of the Plasmodium species. In our Fig 4, we have reported epidemiology of malaria mortality caused by P.falciparum species which as mention is the main cause of cerebral malaria. As such, the fact that we see high frequencies of the CR1 LS haplogroup in the same regional areas where the P. falciparum is linked to high mortality, and it is highly widespread is in agreement with the protective role of the LS against that species. Indeed, cerebral malaria can be caused as well by other species such as P. vivax, but very rarely as compared to falciparum. P. vivax is more widespread in the west areas where we did not detect a high frequency of the CR1 LS haplogroup (Fig 4). On contrary, P. falciparum is highly widespread in the east where we detected high frequencies of the LS haplogroup (Fig 4). As such, we concluded that the main driving force of selection is coming from the P. falciparum species, which is linked to cerebral malaria pathogenesis. For all these reasons, we believe that the methods used in this research is sound and appropriate to identify adaptive variants against of malaria. We mention the cause of cerebral malaria is P. falciparum in discussion (P. 13, L. 275 and P.14, L. 302).

3. Since you inferred that the positive selection already occurred before human migrating out of Africa, I suppose that majority of the extant human racial groups should carry the same polymorphic SNPs which are almost fix, in both endemic and non-endemic extant populations. But your data did not find so. What could be the reason?

Response: We did not infer that positive selection occurred before the out-of-Africa migration, but we stated that the L allele were present before the migration and both L and H alleles have maintained in human populations because there is no selection pressure in either of the alleles. After the out-of-Africa migration and settlement of populations in malaria endemic region, the L allele has become adaptive. Thus, the selection occurred in population-specific manner. To clarify this point, the main text was amended (P. 13, L. 266).

4. Some samples are carrying more variations in the gene. How do you test and remove the mentioned ‘less admixed’ individuals? How do you affirm that the positively selected variants are not due to ancestry / anthropology, instead of natural selection?

Response: 

1) We would like to clarify that we did not remove less admixed populations/individuals, but we removed the highly admixed ones. For example, initially, to select the endemic and non-endemic populations included in this study, we removed highly admixed populations from the GA100K dataset, based on genetic ancestry analysis such as admixture analysis. In addition, for the TMRCA analysis as stated on P. 7, L. 133-135, to build the trees we selected less admixed population groups to have accurate assessment.

2) We believe that the SNPs detected under selection in our study are not due to differences in ancestries since we performed a genome-wide ranking analysis in each population group of similar genetic ancestry. And we compared only within the population groups across whole genome. 

5. Based on the GTEX paper published in 2020, a great majority of 85% of the dataset are of European American. Since European countries are not in the Malaria endemic region, and you also found that there is no signal of selection on CR1 gene among the European samples, you thereby used this public gene expression data (low CR1 expression) as your strong support that infers that this gene is also expressed low among the other multi-racial / ancestries individuals in Asia & Africa. Unless it is tested empirically by any measure of gene expression using RNA from your studies samples, this inference is invalid.

Response: Thanks for indicating the point that the expression data measured by GTEX is based on European populations. We mentioned that the expression data we are referring to was mainly reported for Europeans (P.12, L. 249).

6. The outcome of this work is derived from bioinformatic analyses of individuals with unknown history of malaria infection, and thus the obtained genotypic and allelic frequencies should represent the general polymorphisms in each regional population. However, the conclusion can only be made after a well-designed case-control association test is conducted. As such, your previous work (Gusareva et al., 2021) should have already identified similar results as this current manuscript. I wonder why CR1 analysis was split out from the previous work?

Response: Our previous study indeed mentioned the selection pressure on the CR1 gene. In the present research, we have shown the fine-resolution history of positive selection in the CR1 and identified novel protective variants against malaria in Asians as well as in West Africans that is reported for the first time. Our analysis of the haplogroups (Fig 3 and 4) indicate the age of the protective haplotype and its association with not only the low CR1 expression level but also slow rate of erythrocyte sedimentation. This fine-resolution analysis with genome sequencing dataset can show the population-specific manner of positive selection and convergent evolution especially for understudied Asian populations. This extensive analysis was conducted in follow-up the initial pilot screening reported by Gusareva et al., 2021.

7. The conclusion states that the current findings could be helpful in precision medicine of malaria medical management. Unfortunately, this bioinformatic findings have yet to be extensively tested but the authors already gave strong conclusive statements. I find this misleading and should be removed.

Response: We tone down our conclusions regarding the medical applications of our findings and amended the Conclusions section accordingly (P. 16).

---

## [Decision Letter · Decision Letter 1]

26 Dec 2022

Population-specific positive selection on low CR1 expression in malaria-endemic regions

PONE-D-22-21304R1

Dear Dr. Gusareva,

We’re pleased to inform you that your manuscript has been judged scientifically suitable for publication and will be formally accepted for publication once it meets all outstanding technical requirements.

Kind regards,

Hoh Boon-Peng, PhD

Academic Editor

PLOS ONE

Additional Editor Comments (optional):

Reviewers' comments:

Reviewer's Responses to Questions

**Comments to the Author**

1. If the authors have adequately addressed your comments raised in a previous round of review and you feel that this manuscript is now acceptable for publication, you may indicate that here to bypass the “Comments to the Author” section, enter your conflict of interest statement in the “Confidential to Editor” section, and submit your "Accept" recommendation.

Reviewer #1: All comments have been addressed

Reviewer #2: All comments have been addressed

2. Is the manuscript technically sound, and do the data support the conclusions?

Reviewer #1: Yes

Reviewer #2: Yes

3. Has the statistical analysis been performed appropriately and rigorously? 

Reviewer #1: Yes

Reviewer #2: Yes

4. Have the authors made all data underlying the findings in their manuscript fully available?

Reviewer #1: Yes

Reviewer #2: Yes

5. Is the manuscript presented in an intelligible fashion and written in standard English?

Reviewer #1: Yes

Reviewer #2: Yes

6. Review Comments to the Author

Reviewer #1: (No Response)

Reviewer #2: (No Response)

7. PLOS authors have the option to publish the peer review history of their article (what does this mean?). If published, this will include your full peer review and any attached files.

Reviewer #1: No

Reviewer #2: No

---

## [Editor Report · Acceptance letter]

2 Jan 2023

PONE-D-22-21304R1 

Population-specific positive selection on low CR1 expression in malaria-endemic regions 

Dear Dr. Gusareva:

I'm pleased to inform you that your manuscript has been deemed suitable for publication in PLOS ONE. Congratulations! Your manuscript is now with our production department. 

Kind regards, 

on behalf of

Professor Dr Hoh Boon-Peng 

Academic Editor

PLOS ONE